# Prognostic value of baseline carotid blood flow in critically ill children with septic shock

**Fatma Mamdouh**[1,2◉], **Hafez Bazaraa**[1,2◉], **Ahmed Baz**[3◉], **HebatAllah Fadel Algebaly**[1,2◉]*

**1** Department of Critical Care, Specialized Children Hospital, Cairo University, Cairo, Egypt, **2** Department of Pediatric, Specialized Children Hospital, Cairo University, Cairo, Egypt, **3** Department of Radiology, Kasr Al Aini Hospital, Cairo University, Cairo, Egypt

◉ These authors contributed equally to this work.
* HebatAllah.gebaly@kasralainy.edu.eg

## Abstract

### Background and aim

Hemodynamic monitoring and cardiac output (CO) assessment in the ICU have been trending toward less invasive methods. Carotid blood flow (CBF) was suggested as a candidate for CO assessment. The present study aimed to test the value of carotid artery ultrasound analysis in prediction of mortality in pediatric patients with septic shock.

### Methodology/Principal finding

Forty children with septic shock were included in the study. Upon admission, patients were subjected to careful history taking and thorough clinical examination. The consciousness level was assessed by the Glasgow Coma Scale (GCS). Laboratory assessment included complete blood count, C-reactive protein, arterial blood gases, serum electrolytes, and liver and kidney function tests. Electrical cardiometry was used to evaluate hemodynamic parameters. Patients were also subjected to transthoracic 2-D echocardiography. CBF was evaluated using GE Vivid S5 ultrasound device through dedicated software. At the end of study, 14 patients (35.0%) died. It was found that survivors had significantly higher CBF when compared non-survivors [median (IQR): 166.0 (150.0–187.3) versus 141.0 (112.8–174.3), p = 0.033]. In addition, it was noted that survivors had longer ICU stay when compared with non-survivors [16.5 (9.8–31.5) versus 6.5 (3.0–19.5) days, p = 0.005]. ROC curve analysis showed that CBF could significantly distinguish survivors from non-survivors [AUC (95% CI): 0.3 (0.11–0.48), p = 0.035] (Fig 2). Univariate logistic regression analysis identified type of shock [OR (95% CI): 28.1 (4.9–162.4), p<0.001], CI [OR (95% CI): 0.6 (0.43–0.84), p = 0.003] and CBF [OR (95% CI): 0.98 (0.96–0.99), p = 0.031]. However, in multivariate analysis, only type of shock significantly predicted mortality.

### Conclusions

CBF assessment may be a useful prognostic marker in children with septic shock.

**Data Availability Statement:** All relevant data are within the manuscript and its Supporting Information files.

**Funding:** The authors received no specific funding for this work.

**Competing interests:** The authors have declared that no competing interests exist.

## Introduction

Over the last two decades, hemodynamic monitoring and cardiac output (CO) assessment in the ICU have been trending toward less invasive methods [1]. One of the applications of Point-of-Care Ultrasound (POCUS) is the bedside echocardiogram velocity time interval (VTI) measured through the left ventricular outflow tract is used to measure the CO. Despite its reproducibility, the intensivist/cardiologist can face many challenges in the ICU when dealing with suboptimal cardiac windows due to patient positioning difficulty, mechanical ventilation or wound dressings [2].

In the quest to identify feasible, non-invasive, and reproducible bedside estimates of CO, carotid Doppler imaging shows promise. Two carotid measurements have emerged as potential markers of CO: corrected carotid flow time (CFT) and carotid blood flow (CBF) [3, 4]. CBF measurement has been shown to be feasible to perform at the bedside [5]. Carotid Doppler signal, from a physiological point of view is the proportion of cardiac output that is directed toward the carotid artery. It may vary depending on cerebral blood flow regulation [6].

We hypothesize that the common carotid artery flow can similarly represent an estimate of the CO. This study was designed to test the accuracy, efficiency and feasibility of carotid artery ultrasound analysis in prediction of mortality in pediatric patients with septic shock.

## Patients and methods

### Ethics statement

The study protocol was approved by Cairo university faculty of medicine ethical committee and written consent by the guardians was obtained with approval number I-071017.

### Study participants

This pilot study included 40 children with septic shock from the Pediatric ICU, specialized children hospital, Faculty of medicine, Cairo University. Patients were excluded if they had congenital heart disease, cardiac tamponade, pneumothorax or cardiopulmonary arrest.

### Intervention

Upon admission, patients were subjected to careful history taking and thorough clinical examination. The consciousness level was assessed by the Glasgow Coma Scale (GCS). Laboratory assessment included complete blood count, C-reactive protein, arterial blood gases, serum electrolytes, and liver and kidney function tests.

**Electrical cardiometry.** Used to evaluate hemodynamic parameters e.g. heart rate, CO, cardiac index (CI) and stroke volume (SV). Patients were also subjected to transthoracic 2-D echocardiography.

**Measurement of CBF.** CBF was evaluated using GE Vivid S5 ultrasound device through dedicated software. Assessment of CBF involved obtaining antero-posterior measurements of the common carotid artery diameter in systole within approximately 0.5cm of the common carotid bulb in the long axis with a 12–7 MHz broadband linear array transducer. The common carotid artery was scanned in transverse and longitudinal planes. Spectral Doppler tracings were then obtained by placing a 0.5 mm sample gate through the center of vessel, within 2–3 cm proximal to the carotid bulb in the longitudinal plane, in accordance to standard guidelines [3]. The angle correction cursor was placed parallel to the direction of blood flow. Images with insonation angles >60˚ were excluded because of resultant inaccuracies of flow and velocity measurements at such angles. The VTi is then determined through digitalized Doppler spectral envelopes with the sample obtained at the location that the diameter was

taken. The Doppler gate is placed in the middle of the artery with a 45- to 60-degree angle of insonation. The upper spectral waveform tracing represents peak flow velocity and the lower spectral waveform represents mean flow velocity. Plus signs on the spectral waveform represent peak systolic velocity and end diastolic velocity. CBF was calculated as from the following equation:

CBF = π × (carotid diameter)^2/4 × VTI × heart rate

Where VTI indicates velocity time integral. VTI of the Doppler signal was measured using manual tracings. Intimal-to-intimal carotid diameter was measured at the level of the sample gate.

Studies were performed on ten healthy children before start of the study to evaluate interobserver and intraobserver reproducibility, the radiologist and the intensivist blinded to each other's results alternately performed two measurements on each patient. Intraobserver reproducibility was assessed between the observations by same observer. The intensivist performed the carotid flow scan and all the stored images were reviewed by the expert radiologist.

Considering the pilot nature of the study, we did not perform a priori sample size calculation. The number of patients included in the study was limited by logistic issues.

## Statical analysis

The primary study outcome is patients PICU mortality. Data obtained from the present study were presented as median and interquartile range (IQR) or number and percent. Comparison between numerical data was performed using Mann-Whitney U test while categorical data were compared using Fisher's exact test or chi-square test as appropriate. Receiver operator characteristic (ROC) curve analysis was used to identify capability of CBF to distinguish survivors from non-survivors. Binary logistic regression was used to identify predictors of mortality. All statistical operations were computed using SPSS, 25 (IBM, USA) with p value less than 0.05 considered statistically significant.

## Results

The present study was conducted on 40 children with septic shock. They comprised 13 males (32.5%) and 27 females (67.5%) with a median (IQR) age of 34.5 (19.0–65.0) months.

At the end of study, 14 patients (35.0%) died (Table 1).

### Survivors versus non survivors

Comparison between survivors and non-survivors regarding the studied variables revealed that survivors had significantly higher frequency of warm shock (88.5% versus 21.4%, p<0.001). Non-survivors had significantly higher ALT and AST levels. In addition, it was found that non-survivors had significantly lower EF %, FS %, SV, CI, and higher CVP when compared with survivors. It was also found that survivors had significantly higher CBF when compared non-survivors [median (IQR): 166.0 (150.0–187.3) versus 141.0 (112.8–174.3), p = 0.033] (Fig 1). In addition, it was noted that survivors had longer ICU stay when compared with non-survivors [16.5 (9.8–31.5) versus 6.5 (3.0–19.5) days, p = 0.005]. clinical, laboratory and therapeutic parameters of the studied groups are shown in Table 1.

### Warm versus cold shock

Interestingly, patients with warm shock had significantly higher CBF when compared with those with cold shock [median (IQR): 168.5 (150.0–188.5) versus 141.0 (122.0–164.5), p<0.001] (Fig 2).

**Table 1. Clinical data and therapeutic interventions in the studied patients.**

| | All patients | Survivors | Non-survivors | P value |
|---|---|---|---|---|
| | N = 40 | n = 26 | n = 14 | |
| **Age** (months) | 34.5 (19.0–65.0) | 25.5 (18.8–65.0) | 60.0 (19.3–68.3) | 0.35 |
| **Male/female** n | 13/27 | 6/20 | 7/7 | 0.083 |
| **Weight** (Kg) | 14.5 (11.0–17.0) | 12.0 (11.0–16.3) | 16.4 (11.0–18.3) | 0.2 |
| **Body surface area** (m^2) | 0.63 (0.5–0.7) | 0.54 (0.5–0.68) | 0.68 (0.5–0.74) | 0.2 |
| **Associated comorbidities** n (%) | | | | |
| Neurological | 4 (10.0) | 3 (11.5) | 1 (7.1) | 0.66 |
| Cardiac | 5 (12.5) | 1 (3.8) | 4 (28.6) | 0.024 |
| Respiratory | 17 (42.5) | 14 (53.8) | 3 (21.4) | 0.048 |
| Gastrointestinal | 7 (17.5) | 4 (15.4) | 3 (21.4) | 0.63 |
| Others | 7 (17.5) | 4 (15.4) | 3 (21.4) | 0.63 |
| **Type of shock** n (%) | | | | |
| Warm | 26 (65.0) | 23 (88.5) | 3 (21.4) | <0.001 |
| Cold | 14 (35.0) | 3 (11.5) | 11 (78.6) | |
| **GCS**[*] median (IQR) | 6.0 (5.3–7.0) | 7.0 (6.0–7.0) | 5.5 (5.0–7.0) | 0.071 |
| **Clinical data** median (IQR) | | | | |
| Temperature (˚C) | 37.0 (36.3–37.5) | 37.0 (36.8–37.9) | 36.3 (36.0–37.0) | 0.008 |
| Heart rate (beat/m.) | 165.0 (156.0–179.3) | 165.0 (157.3–180.0) | 161.5 (155.8–173.5) | 0.31 |
| MAP[*] (mmHg) | 67.5 (62.0–78.0) | 65.5 (54.0–75.8) | 70.5 (65.3–88.5) | 0.14 |
| Respiratory rate (breath/m.) | 49.0 (41.3–60.0) | 53.0 (43.0–60.0) | 44.0 (35.0–52.5) | 0.076 |
| UOP [*](ml/m^2/day) | 1.0 (0.8–1.8) | 1.0 (0.8–1.8) | 0.9 (0.4–1.3) | 0.46 |
| **Laboratory data** median (IQR) | | | | |
| Hb (gm/dl) | 8.5 (7.3–10.2) | 9.1 (7.3–10.3) | 8.5 (7.3–10.1) | 0.9 |
| Platelets (×10^3/ml) | 335.0 (124.5–462.0) | 350.0 (201.0–534.8) | 128.5 (64.5–456.0) | 0.067 |
| WBCs[*] (×10^3/ml) | 15.8 (8.7–20.3) | 16.7 (12.9–20.4) | 13.1 (4.1–20.3) | 0.31 |
| TSB [*](mg/dl) | 2.0 (1.5–3.5) | 2.0 (1.3–3.0) | 3.0 (1.8–4.6) | 0.12 |
| AST[*] (U/L) | 74.0 (38.5–96.0) | 47.5 (33.0–80.3) | 93.0 (75.8–295.0) | 0.001 |
| ALT [*](U/L) | 66.0 (45.0–117.5) | 51.0 (33.3–76.0) | 112.0 (64.3–384.8) | 0.001 |
| Creatinine (mg/dl) | 0.8 (0.6–1.1) | 0.8 (0.6–1.0) | 1.0 (0.6–1.4) | 0.42 |
| Urea (mg/dl) | 71.0 (56.8–98.8) | 68.0 (58.3–90.3) | 84.5 (54.5–114.3) | 0.47 |
| RBS[*] (mg/dl) | 114.5 (54.8–187.0) | 106.0 (56.3–162.5) | 138.5 (46.8–235.0) | 0.55 |
| Na (mEq/L) | 143.0 (134.0–149.8) | 143.0 (134.8–150.0) | 137.0 (132.8–143.5) | 0.13 |
| K (mEq/L) | 3.8 (2.9–4.7) | 3.8 (2.8–4.4) | 4.6 (3.1–4.9) | 0.19 |
| pH | 7.3 (7.2–7.3) | 7.3 (7.2–7.3) | 7.3 (7.2–7.3) | 0.81 |
| HCO3 (mEq/L) | 15.4 (13.8–16.4) | 15.5 (14.0–16.5) | 15.3 (13.4–16.3) | 0.71 |
| PaO2 (mmHg) | 100.0 (70.0–100.0) | 100.0 (73.8–100.0) | 70.0 (63.0–100.0) | 0.17 |
| PaCo2 (mmHg) | 25.5 (21.0–46.8) | 25.5 (20.0–46.3) | 25.0 (21.8–49.5) | 0.86 |
| FiO2 (%) | 47.0 (40.0–100.0) | 42.5 (40.0–100.0) | 100.0 (43.8–100.0) | 0.14 |
| CRP[*] (mg/dl) | 133.5 (113.0–228.8) | 144.0 (109.8–216.3) | 133.0 (117.8–305.3) | 0.99 |
| **Echocardiographic findings** median (IQR) | | | | |
| EF [*](%) | 60.5 (34.0–68.8) | 62.5 (57.8–69.0) | 34.0 (32.5–44.5) | 0.004 |
| FS[*] (%) | 37.5 (22.3–39.8) | 39.0 (36.5–40.5) | 22.5 (20.0–27.0) | 0.001 |
| SV [*](ml) | 18.5 (11.9–22.7) | 19.8 (17.5–23.5) | 11.9 (7.9–14.1) | 0.001 |
| CI [*](L/min./m^2) | 6.2 (2.6–7.4) | 6.4 (6.1–7.7) | 2.6 (2.3–4.0) | 0.002 |
| CVP [*](cmH2O) | 3.5 (-1.5–6.0) | 3.0 (-2.0–5.0) | 6.0 (1.8–8.5) | 0.019 |
| **Carotid blood flow** | 161.5 (139.3–187.0) | 166.0 (150.0–187.3) | 141.0 (112.8–174.3) | 0.033 |
| **Therapeutic interventions** n (%) | | | | |

*(Continued)*

**Table 1.** (Continued)

|  | All patients | Survivors | Non-survivors | P value |
|---|---|---|---|---|
|  | N = 40 | n = 26 | n = 14 |  |
| Mechanical ventilation | 33 (82.5) | 20 (76.9) | 13 (92.9) | 0.21 |
| Blood transfusion | 22 (55.0) | 14 (53.9) | 8 (57.1) | 0.84 |
| Colloid infusion | 7 (17.5) | 4 (15.4) | 3 (21.4) | 0.63 |
| Crystalloid infusion | 35 (87.5) | 25 (96.2) | 10 (71.4) | 0.024 |
| Adrenaline | 27 (67.5) | 18 (69.2) | 9 (64.3) | 0.75 |
| Noradrenaline | 24 (60.0) | 22 (84.6) | 2 (14.3) | <0.001 |
| Dobutamine | 11 (27.5) | 2 (7.7) | 9 (64.3) | <0.001 |
| **ICU stay** (days) | 14.0 (7.0–21.8) | 16.5 (9.8–31.5) | 6.5 (3.0–19.5) | 0.005 |

ALT, alanine transaminase; AST, aspartate transaminase; CI, cardiac index; CVP, central venous pressure; CRP, C reactive protein, GCS, Glasgow coma score; EF, ejection fraction; FS, fractional shortening; MAP, mean blood pressure; SV, stroke volume; RBS, random blood sugar; TSB, total serum bilirubin; WBCs, white blood cells.

### Carotid blood flow

Correlation analysis revealed modest significant correlation between CBF and age (r = 0.42, p = 0.008), weight (r = 0.38, p = 0.017), surface area (r = 0.38, p = 0.017) and SV (r = .49, p = 0.001) (Table 2).

### Mortality & CBF

ROC curve analysis showed that CBF could significantly distinguish survivors from non-survivors [AUC (95% CI): 0.71 (0.52–0.89), p = 0.035], sensitivity:71.4%, specificity:53.8% (Fig 3).

Univariate logistic regression analysis identified type of shock [OR (95% CI): 28.1 (4.9–162.4), p<0.001], CI [OR (95% CI): 0.6 (0.43–0.84), p = 0.003] and CBF [OR (95% CI): 0.98

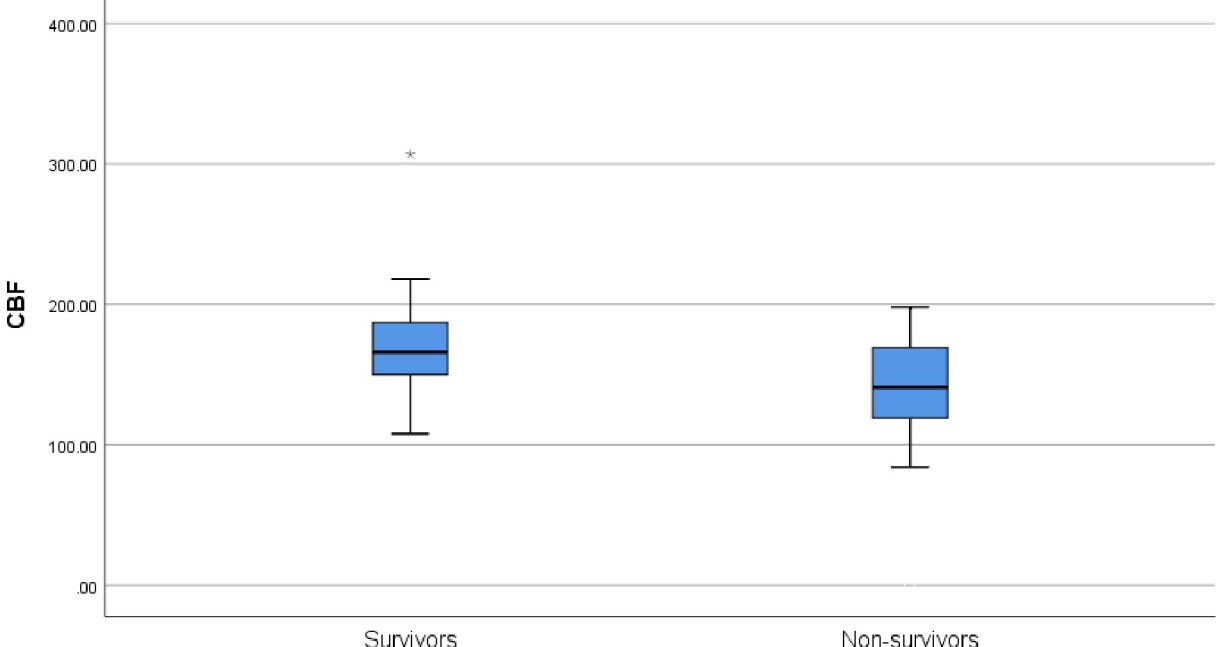

**Fig 1. Difference in CBF between survivors and non survivors.**

**Table 2. Correlation between CBF and various clinical and laboratory parameters.**

| | Carotid Artery Blood Flow | |
| --- | --- | --- |
| | **R** | **p** |
| **Age** | 0.42 | 0.008 |
| **Weight** | 0.38 | 0.017 |
| **Body surface area** | 0.38 | 0.017 |
| GCS * | 0.07 | 0.65 |
| Temperature | 0.37 | 0.2 |
| Heart rate | 0.2 | 0.22 |
| MAP* | -0.13 | 0.43 |
| Respiratory rate | -0.004 | 0.98 |
| UOP* | 0.21 | 0.19 |
| Hb | -0.11 | 0.49 |
| Platelets | 0.19 | 0.24 |
| WBCs* | 0.08 | 0.64 |
| TSB* | -0.1 | 0.56 |
| AST* | 0.008 | 0.96 |
| ALT* | -0.04 | 0.92 |
| Creatinine | 0.26 | 0.1 |
| Urea | 0.28 | 0.08 |
| RBS* | -0.01 | 0.95 |
| Na | 0.12 | 0.47 |
| K | 0.14 | 0.39 |
| pH | -0.26 | 0.11 |
| **HCO3** | -0.46 | 0.003* |
| PaO2 | -0.04 | 0.79 |
| PaCo2 | -0.09 | 0.59 |
| FiO2 | -0.15 | 0.35 |
| CRP* | 0.06 | 0.72 |
| EF* | 0.08 | 0.64 |
| FS* | 0.28 | 0.08 |
| **SV*** | 0.49 | 0.001* |
| CI* | 0.22 | 0.18 |
| CVP* | -0.16 | 0.33 |
| ICU stay* | 0.22 | 0.17 |

ALT, alanine transaminase; AST, aspartate transaminase; CI, cardiac index; CVP, central venous pressure; CRP, C reactive protein, GCS, Glasgow coma score; EF, ejection fraction; FS, fractional shortening; ICU, intensive care unit; MAP, mean blood pressure; SV, stroke volume; RBS, random blood sugar; TSB, total serum bilirubin; UOP, urine output; WBCs, white blood cells.

(0.96–0.99), p = 0.031]. However, in multivariate analysis, only type of shock significantly predicted mortality (Table 3).

## Discussion

Septic shock is a clinically devastating condition. In the United States, severe sepsis/septic shock accounts for over annual 75,000 pediatric hospitalizations [7, 8]. Assessment of severity of illness at admission is important for effective patient management, prognostication, and optimum utilization of resources [9]. Simple interventions such as early rapid fluid

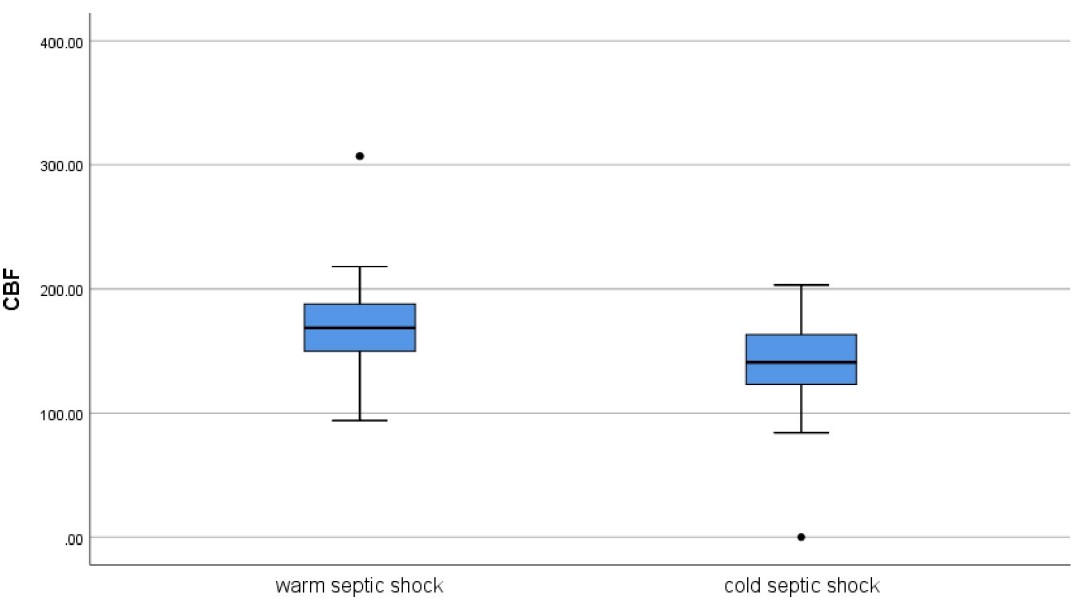

**Fig 2. Difference in CBF between warm and cold septic shock.**

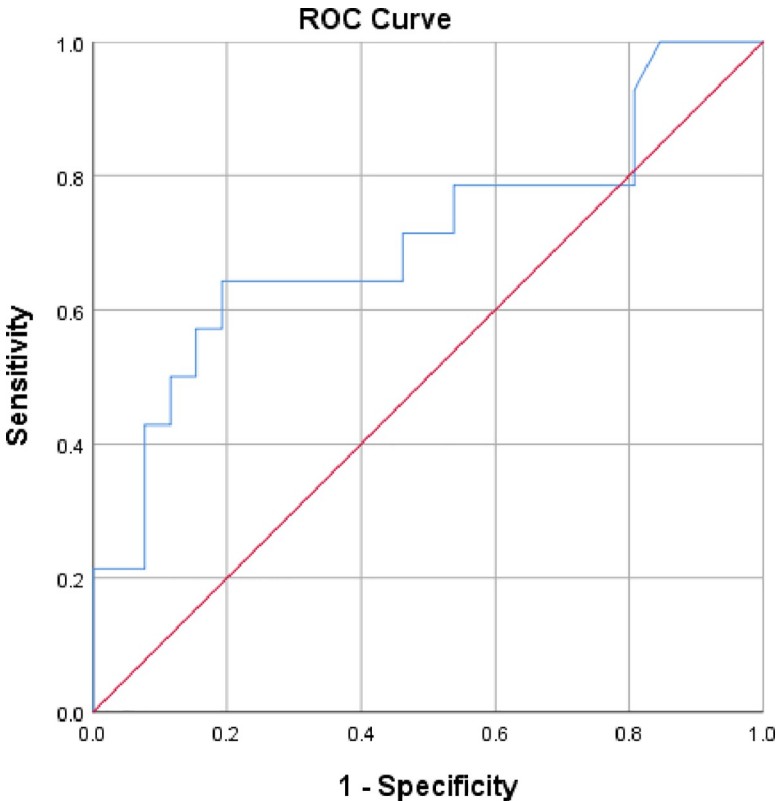

**Fig 3. ROC curve for sensitivity and specificity of CBF for mortality.**

**Table 3. Predictors of mortality in the studied patients.**

|  | Univariate analysis | | | Multivariate analysis | | |
|---|---|---|---|---|---|---|
|  | OR | 95%CI | P | OR | 95%CI | P |
| **Age** | 1.02 | 0.99–1.05 | 0.19 | - | - | - |
| **Sex** | 0.3 | 0.08–1.02 | 0.09 | - | - | - |
| **Type of shock** | 28.1 | 4.9–162.4 | <0.001 | 116.8 | 1.69–8101.7 | 0.028 |
| **CI*** | 0.6 | 0.43–0.84 | 0.003 | 1.36 | 0.71–2.62 | 0.36 |
| **CBF*** | 0.98 | 0.96–0.99 | 0.031 | 0.99 | 0.95–1.01 | 0.3 |

CI, cardiac index; CBF, cerebral blood flow

administration, early antibiotics therapy, oxygen supplementation, and early use of inotropes have shown to improve the outcome [10]. We conducted this study to explore the role of admission carotid blood flow assessment as noninvasive prognostic tool in pediatric patients with septic shock.

In our study, the mortality rate was 40.0% among the studied 40 patients. These figure is notably higher than that reported by some studies [11, 12] but it is also lower that morality rates reported by other studies [13–15]. Discrepancy between various studies is probably attributed to the different nature and severity of underlying illnesses affecting patients recruited by different studies.

Interestingly, the present study found significantly higher carotid blood flow CBF in survivors as compared to non-survivors. Moreover ROC curve analysis revealed that could significantly distinguish survivors from non-survivors. In addition, univariate logistic analysis identified CBF as a significant predictor of PICU mortality. However, in multivariate analysis, CBF couldn't predict PICU mortality in the studied patients.

To the best of our knowledge it is the first study to use carotid artery blood flow as hemodynamic parameter in prediction of prognosis of children with septic shock, despite the presence of few adult studies [3, 16]. We think that The use point-of-care ultrasound to estimate the carotid blood flow is rapid, easy to learn and noninvasive method for prognostic evaluation in critically unstable children who may not tolerate any invasive maneuver. By having user-friendly, accurate and less time-consuming non-invasive hemodynamic monitoring methods, suitable interventions would result in less complications, morbidity and mortality [17].

The Surviving Sepsis Campaign International Guidelines for the Management of Septic Shock in Children 2020 reported that distinction between cold and warm shock if advanced hemodynamic monitoring is available may be helpful to assess patient physiology and direct inotropes and vasopressor therapy [18]. We could observe that the carotid blood flow is less in patients with cold compared to warm shock type. However, the higher SVRI in cold shock could explain the reduction of carotid blood flow and whether that fluid augmentation and aiming at normalization of SVRI could influence the outcome needs to be explored.

## Study limitation

First, our study is a single center, with small number of septic shock patients. Second, we did not observe if the changes in the cerebral autoregulation in pediatric septic shock could impact the carotid blood flow. Also, whether targeting increase carotid blood flow will improve survival or not needs further study.

## Supporting information

**S1 File. Data of patients investigated for carotid blood flow.**
(XLS)

## Acknowledgments

We would like to thank the PICU team without their sincere work we would not accomplish this work.

## Author Contributions

**Conceptualization:** HebatAllah Fadel Algebaly.

**Data curation:** Fatma Mamdouh, Ahmed Baz.

**Formal analysis:** Fatma Mamdouh, Hafez Bazaraa.

**Methodology:** Hafez Bazaraa, Ahmed Baz, HebatAllah Fadel Algebaly.

**Supervision:** Hafez Bazaraa.

**Visualization:** HebatAllah Fadel Algebaly.

**Writing – original draft:** Ahmed Baz.

**Writing – review & editing:** Ahmed Baz, HebatAllah Fadel Algebaly.

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
