## [Decision Letter · Decision Letter 0]

11 Jan 2021

PONE-D-20-35629

Prognostic Value of Baseline Carotid Blood Flow in Critically Ill Children with Septic Shock

PLOS ONE

Dear Dr. Algebaly,

Thank you for submitting your manuscript to PLOS ONE. After careful consideration, we feel that it has merit but does not fully meet PLOS ONE’s publication criteria as it currently stands. Therefore, we invite you to submit a revised version of the manuscript that addresses the points raised during the review process.

All issues raised by reviewers are required.

We look forward to receiving your revised manuscript.

Kind regards,

Vincenzo Lionetti, M.D., PhD

Academic Editor

PLOS ONE

Journal Requirements:

2. We noticed you have some minor occurrence of overlapping text with the following previous publications, which needs to be addressed:

https://rd.springer.com/article/10.1007/s40477-014-0139-9?code=897273d4-47ee-4cff-8464-94a130407a9e

https://westjem.com/videos/can-emergency-physicians-perform-common-carotid-doppler-flow-measurements-to-assess-volume-responsiveness.html

https://theultrasoundjournal.springeropen.com/articles/10.1186/s13089-017-0065-0

3. Please ensure you have discussed any potential limitations of your study in the Discussion, including study design, sample size and/or potential confounders.

4. Please provide a sample size and power calculation in the Methods, or discuss the reasons for not performing one before study initiation.

5. Please provide additional details regarding participant consent.

In the ethics statement in the Methods and online submission information, please ensure that you have specified what type you obtained (for instance, written or verbal, and if verbal, how it was documented and witnessed).

6. Thank you for including your ethics statement: 

"I-071017

Ethical committee of cairo university, faculty of medicine

written consent was obtained from parents".   

a. Please amend your current ethics statement to confirm that your named institutional review board or ethics committee specifically approved this study.

7. Please include a separate caption for each figure in your manuscript.

Reviewers' comments:

Reviewer's Responses to Questions

**Comments to the Author**

1. Is the manuscript technically sound, and do the data support the conclusions?

Reviewer #1: Yes

Reviewer #2: Partly

2. Has the statistical analysis been performed appropriately and rigorously? 

Reviewer #1: Yes

Reviewer #2: Yes

3. Have the authors made all data underlying the findings in their manuscript fully available?

Reviewer #1: Yes

Reviewer #2: No

4. Is the manuscript presented in an intelligible fashion and written in standard English?

Reviewer #1: Yes

Reviewer #2: Yes

5. Review Comments to the Author

Reviewer #1: The evaluation of the haemodynamic status in critical patient with septic shock has an important prognostic value. An easy-to-measure indicator helps to monitor the unstable patient admitted to the ICU.

The manuscript is well written and describes a simple and useful method for monitoring cardiac output in critical patients

The statistical analysis is well done and the data reported are exhaustive.

The language in submitted articles is clear, correct, and easy to understand without errors.

The data presented in the manuscript support the final conclusions.

Reviewer #2: The authors evaluated the value of carotid artery ultrasound analysis in prediction of mortality in paediatric patients with septic shock.

For this purpose, they enrolled forty children with septic shock and concluded that carotid blood flow (CBF) assessment may be a useful prognostic marker in this setting.

Specific comments:

- The changes in CBF reported in this analysis might simply reflect the blood flow redistribution typical of septic shock. In fact, the changes were larger in cold than in warm shock. Hence, the reduction of CBF might be a feature of more advanced shock. The authors should comment on this: what is their view on this point? What is the potential clinical usefulness of the index? Should it just reflect advanced disease, would it have an actual impact on patient management?

- Which is the additive prognostic value of CBF over the traditional clinical parameters of hemodynamic impairment and/or hypo-perfusion of peripheral organs and tissues (lactate, indices of multi-organ failure, CVP)?

- ROC Curve in Figure 2 does not match with results provided in Results Section [AUC (95% CI): 0.3 (0.11 – 0.48), p=0.035]. Please check.

- In the Results Section the authors stated: “Correlation analysis revealed significant correlation between CBF and age (r=0.42, p=0.008), weight (r=0.38, p=0.017), surface area (r=0.38, p=0.017) and SV (r=0.49, p=0.001)”. Correlations are indeed significant, but modest according to the “r” coefficients.

6. PLOS authors have the option to publish the peer review history of their article (what does this mean?). If published, this will include your full peer review and any attached files.

Reviewer #1: No

Reviewer #2: No

---

## [Author Response · Author response to Decision Letter 0]

19 Feb 2021

Response to reviewers

Dear Sir,

It was great pleasure to allow me to improve the manuscript.

Please ensure that your manuscript meets PLOS ONE's style requirements, including those for file naming. The PLOS ONE style templates :

Reformatting according to PLOS ONE requirements.

2. We noticed you have some minor occurrence of overlapping text with the following previous publications, which needs to be addressed:

https://rd.springer.com/article/10.1007/s40477-014-0139-9?code=897273d4-47ee-4cff-8464-94a130407a9e

https://westjem.com/videos/can-emergency-physicians-perform-common-carotid-doppler-flow-measurements-to-assess-volume-responsiveness.html

https://theultrasoundjournal.springeropen.com/articles/10.1186/s13089-017-0065-0

Introduction was modified to avoid the overlapping text with previous publications to 

3. Please ensure you have discussed any potential limitations of your study in the Discussion, including study design, sample size and/or potential confounders.

Study limitations section was added. Lines 188-191

 first, our study is a single center , with small number of septic shock patients. Second, we did not observe if the changes in the cerebral autoregulation in pediatric septic shock could impact the carotid blood flow. Also, whether targeting increase carotid blood flow will improve survival or not needs further study.

4.Please provide a sample size and power calculation in the Methods, or discuss the reasons for not performing one before study initiation.

Considering the pilot nature of the study, we did not perform a priori sample size calculation. The number of patients included in the study was limited by logistic issues. Lines 85-86

4.Please provide additional details regarding participant consent.In the ethics statement in the Methods and online submission information, please ensure that you have specified what type you obtained (for instance, written or verbal, and if verbal, how it was documented and witnessed).

 The study protocol was approved by Cairo university faculty of medicine ethical committee with written consent with approval number "I-071017. Line 52-53

5.Please include a separate caption for each figure in your manuscript.

Attached after the corresponding paragraph. Line 124 

 Line 142.

Reviewer #2: The authors evaluated the value of carotid artery ultrasound analysis in prediction of mortality in paediatric patients with septic shock.

Specific comments:

- The changes in CBF reported in this analysis might simply reflect the blood flow redistribution typical of septic shock. In fact, the changes were larger in cold than in warm shock. Hence, the reduction of CBF might be a feature of more advanced shock. The authors should comment on this: what is their view on this point? What is the potential clinical usefulness of the index? Should it just reflect advanced disease, would it have an actual impact on patient management?

Lines 180-187: The Surviving Sepsis Campaign International Guidelines for the Management of Septic Shock in Children 2020 reported that distinction between cold and warm shock if advanced hemodynamic monitoring is available may be helpful to assess patient physiology and direct inotropes and vasopressor therapy. We could observe that the carotid blood flow is less in patients with cold compared to warm shock type. However, the higher SVRI in cold shock could explain the reduction of carotid blood flow and whether that fluid augmentation and aiming at normalization of SVRI could influence the outcome needs to be explored.

- Which is the additive prognostic value of CBF over the traditional clinical parameters of hemodynamic impairment and/or hypo-perfusion of peripheral organs and tissues (lactate, indices of multi-organ failure, CVP)?

Line 167-168: The use point-of-care ultrasound to estimate the carotid blood flow is a rapid, easy to learn and noninvasive method for prognostic evaluation in critically unstable children who may not tolerate any invasive maneuver. 

- ROC Curve in Figure 2 does not match with results provided in Results Section [AUC (95% CI): 0.3 (0.11 – 0.48), p=0.035]. Please check.

Line 140: [AUC (95% CI): 0.71 (0.52 – 0.89), p=0.035], sensitivity:71.4%, specifity:53.8% 

- In the Results Section the authors stated: “Correlation analysis revealed significant correlation between CBF and age (r=0.42, p=0.008), weight (r=0.38, p=0.017), surface area (r=0.38, p=0.017) and SV (r=0.49, p=0.001)”. Correlations are indeed significant, but modest according to the “r” coefficients.

Line 126: Correlation analysis revealed modest significant correlation between CBF and age (r=0.42, p=0.008

---

## [Decision Letter · Decision Letter 1]

18 Mar 2021

PONE-D-20-35629R1

Prognostic Value of Baseline Carotid Blood Flow in Critically Ill Children with Septic Shock

PLOS ONE

Dear Dr. Algebaly,

Thank you for submitting your manuscript to PLOS ONE. After careful consideration, we feel that it has merit but does not fully meet PLOS ONE’s publication criteria as it currently stands. Therefore, we invite you to submit a revised version of the manuscript that addresses the points raised during the review process.

ACADEMIC EDITOR: All issues highlighted during revision are required.

We look forward to receiving your revised manuscript.

Kind regards,

Vincenzo Lionetti, M.D., PhD

Academic Editor

PLOS ONE

Journal Requirements:

Reviewers' comments:

Reviewer's Responses to Questions

**Comments to the Author**

1. If the authors have adequately addressed your comments raised in a previous round of review and you feel that this manuscript is now acceptable for publication, you may indicate that here to bypass the “Comments to the Author” section, enter your conflict of interest statement in the “Confidential to Editor” section, and submit your "Accept" recommendation.

Reviewer #1: All comments have been addressed

Reviewer #2: (No Response)

2. Is the manuscript technically sound, and do the data support the conclusions?

Reviewer #1: (No Response)

Reviewer #2: Yes

3. Has the statistical analysis been performed appropriately and rigorously? 

Reviewer #1: (No Response)

Reviewer #2: Yes

4. Have the authors made all data underlying the findings in their manuscript fully available?

Reviewer #1: (No Response)

Reviewer #2: No

5. Is the manuscript presented in an intelligible fashion and written in standard English?

Reviewer #1: (No Response)

Reviewer #2: Yes

6. Review Comments to the Author

Reviewer #1: (No Response)

Reviewer #2: I am happy with the changes and the authors' response.

Some minor points still to address:

- Can you please provide inter and intra-observer reproducibilty data of CBF?

- Please provide a figure showing different patterns of CBF in different groups.

7. PLOS authors have the option to publish the peer review history of their article (what does this mean?). If published, this will include your full peer review and any attached files.

Reviewer #1: No

Reviewer #2: No

---

## [Author Response · Author response to Decision Letter 1]

10 Apr 2021

Dear Sir,

It was great pleasure to allow me to improve the manuscript.

Reference list was revised and non of the papers were retracted 

Reference 12 was missing and I added it to the reference section.

Reviewer #2: I am happy with the changes and the authors' response.

Some minor points still to address:

- Can you please provide inter and intra-observer reproducibilty data of CBF?

- Please provide a figure showing different patterns of CBF in different groups.

Studies were performed on ten healthy children before start of the study to evaluate interobserver and intraobserver reproducibility, the radiologist and the intensivist blinded to each other's results alternately performed two measurements on each patient. Intraobserver reproducibility was assessed between the observations by same observer. The intensivist performed the carotid flow scan and all the stored images were reviewed by the expert radiologist.

Figure 1 about subgroups:

---

## [Decision Letter · Decision Letter 2]

21 Apr 2021

Prognostic Value of Baseline Carotid Blood Flow in Critically Ill Children with Septic Shock

PONE-D-20-35629R2

Dear Dr. Algebaly,

We’re pleased to inform you that your manuscript has been judged scientifically suitable for publication and will be formally accepted for publication once it meets all outstanding technical requirements.

Kind regards,

Vincenzo Lionetti, M.D., PhD

Academic Editor

PLOS ONE

Additional Editor Comments (optional):

Reviewers' comments:

Reviewer's Responses to Questions

**Comments to the Author**

1. If the authors have adequately addressed your comments raised in a previous round of review and you feel that this manuscript is now acceptable for publication, you may indicate that here to bypass the “Comments to the Author” section, enter your conflict of interest statement in the “Confidential to Editor” section, and submit your "Accept" recommendation.

Reviewer #1: All comments have been addressed

2. Is the manuscript technically sound, and do the data support the conclusions?

Reviewer #1: Yes

3. Has the statistical analysis been performed appropriately and rigorously? 

Reviewer #1: Yes

4. Have the authors made all data underlying the findings in their manuscript fully available?

Reviewer #1: Yes

5. Is the manuscript presented in an intelligible fashion and written in standard English?

Reviewer #1: Yes

6. Review Comments to the Author

Reviewer #1: (No Response)

7. PLOS authors have the option to publish the peer review history of their article (what does this mean?). If published, this will include your full peer review and any attached files.

Reviewer #1: No

---

## [Editor Report · Acceptance letter]

24 Jun 2021

PONE-D-20-35629R2 

Prognostic Value of Baseline Carotid Blood Flow in Critically Ill Children with Septic Shock 

Dear Dr. Algebaly:

I'm pleased to inform you that your manuscript has been deemed suitable for publication in PLOS ONE. Congratulations! Your manuscript is now with our production department. 

Kind regards, 

on behalf of

Prof. Vincenzo Lionetti 

Academic Editor

PLOS ONE